# The representational space of observed actions

**Raffaele Tucciarelli[1], Moritz Wurm[2], Elisa Baccolo[2], Angelika Lingnau[1,2,3]***

[1]Department of Psychology, Royal Holloway University of London, Egham, United Kingdom; [2]Center for Mind/Brain Sciences (CIMeC), University of Trento, Rovereto, Italy; [3]Institute of Psychology, University of Regensburg, Regensburg, Germany

**Abstract** Categorizing and understanding other people's actions is a key human capability. Whereas there exists a growing literature regarding the organization of objects, the representational space underlying the organization of observed actions remains largely unexplored. Here we examined the organizing principles of a large set of actions and the corresponding neural representations. Using multiple regression representational similarity analysis of fMRI data, in which we accounted for variability due to major action components (body parts, scenes, movements, objects, sociality, transitivity) and three control models (distance between observer and actor, number of people, HMAX-C1), we found that the semantic dissimilarity structure was best captured by patterns of activation in the lateral occipitotemporal cortex (LOTC). Together, our results demonstrate that the organization of observed actions in the LOTC resembles the organizing principles used by participants to classify actions behaviorally, in line with the view that this region is crucial for accessing the meaning of actions.

## Introduction

Humans can perform and recognize a striking number of different types of actions, from hammering a nail to performing open heart surgery. However, most of what we know about the way we control and recognize actions is based on a rich literature on prehension movements in humans and non-human primates. This literature revealed a widespread network of fronto-parietal regions, with a preference along the dorso-medial stream and the dorso-lateral stream for reaching and grasping movements, respectively (for reviews see e.g. *Culham and Valyear, 2006*; *Rizzolatti and Matelli, 2003*; *Turella and Lingnau, 2014*). Less is known regarding the organization of more complex actions (for exceptions, see *Abdollahi et al., 2013*; *Jastorff et al., 2010*; *Wurm et al., 2017*). According to which principles are different types of actions organized in the brain, and do these principles help us understand how we are able to tell that two actions, for example running and riding a bike, are more similar to each other than two other actions, for example riding a bike and reading a book? Are observed actions that we encounter on a regular basis organized according to higher-level semantic categorical distinctions (e.g. between locomotion, object manipulation, communication actions) and further overarching organizational dimensions? Note that higher-level semantic categories often covary with more basic action components (such as body parts, movement kinematics, and objects) of perceived action scenes. As an example, locomotion actions tend to involve the legs, consist of repetitive movements, can involve vehicles such as a bike and often take place outdoors, whereas communicative actions often involve mouth/lip movements, consist of small movements, can involve objects such as a mobile phone, and can take place in a variety of different scenes. Disentangling these levels neurally presents an analytical challenge that has not been addressed so far.

A number of recent studies used multivariate pattern analysis (MVPA) (*Haxby et al., 2001*) to examine which brain areas are capable to distinguish between different observed actions (e.g.

*For correspondence:
angelika.lingnau@psychologie.uni-regensburg.de

Competing interests: The authors declare that no competing interests exist.

opening vs closing, slapping vs lifting an object, or cutting vs peeling) (*Wurm et al., 2017*; *Wurm et al., 2016*; *Wurm and Lingnau, 2015*; *Hafri et al., 2017*; *Oosterhof et al., 2010*; *Oosterhof et al., 2012*). The general results that emerged from these studies is that it is possible to distinguish between different actions on the basis of patterns of brain activation in the lateral occipito-temporal cortex (LOTC), the inferior parietal lobe (IPL) and the ventral premotor cortex (PMv). In line with this view, LOTC has been shown to contain action-related object properties (*Bracci and Peelen, 2013*). LOTC and IPL, but not the PMv, furthermore showed a generalization across the way in which these actions were performed (e.g. performing the same action with different kinematics), suggesting that these areas represent actions at more general levels and thus possibly the meaning of the actions (*Wurm and Lingnau, 2015*). However, previous studies could not unambiguously determine what kind of information was captured from observed actions: movement trajectories and body postures (*Wurm et al., 2017*; *Oosterhof et al., 2010*; *Oosterhof et al., 2012*), certain action precursors at high levels of generality (*Wurm and Lingnau, 2015*) (e.g. object state change), or more complex semantic aspects that go beyond the basic constituents of perceived actions and that represent the meaning of actions at higher integratory levels. In the latter case, the LOTC and the IPL should also reflect the semantic similarity structure of a wide range of actions: Running shares more semantic aspects with riding a bike than with reading; therefore, semantic representations of running and riding a bike should be more similar with each other than with the semantic representation of reading.

To determine the structure of the similarity in meaning (which we will refer to as *semantic similarity* in the remainder of this paper), we used inverse multidimensional scaling (MDS) (*Kriegeskorte and Mur, 2012*) of a range of different actions (*Figure 1*). To test the prediction that the LOTC and the IPL reflect the semantic similarity structure determined behaviorally (using inverse MDS), we carried out an fMRI study in the same group of participants. To control for action components that often covary with action semantics, we carried out inverse MDS in the same group of participants for the similarity of actions with respect to (a) the body parts involved in the action (body model), (b) the scenes in which these actions typically take place (scene model), (c) movement kinematics involved in the actions (movement model), and (d) objects involved in these actions (object model).

To be able to relate our results to a previous study (*Wurm et al., 2017*) explicitly comparing actions directed towards an object and actions directed towards another person, we included two additional models capturing the *transitivity* and *sociality* of the actions, respectively (see Materials and methods, section *Construction of Representational Dissimilarity Matrices*, for details). Moreover, to account for the fact that some of the images depicting actions were photographed at a shorter distance to the actor(s) than other images, and since some actions included two actors, whereas the majority of actions depicted one actor only, we included additional models capturing the distance to the actor (near, medium, far) and the number of actors involved in the action (one vs two; see Materials and methods, section *Construction of Representational Dissimilarity Matrices* for details). Finally, to be able to rule out that any differences obtained in the RSA are not due to low-level differences between the actions, we included the second level (C1) of the HMAX model (*Riesenhuber and Poggio, 1999*; *Serre et al., 2007*) (see Methods, section *Construction of Representational Dissimilarity Matrices*, for details).

In the fMRI study, we examined which brain regions capture the semantic similarity structure determined in the behavioral experiment, using representational similarity analysis (*Kriegeskorte et al., 2008a*) (RSA). Moreover, to examine which brain areas capture semantic similarity over and beyond the action components described above, and to control for potential confounds and low level differences in the stimulus material, we carried out a multiple regression RSA for each of the models while accounting for all the remaining models (see Materials and methods, section *Representational Similarity Analysis*, for details).

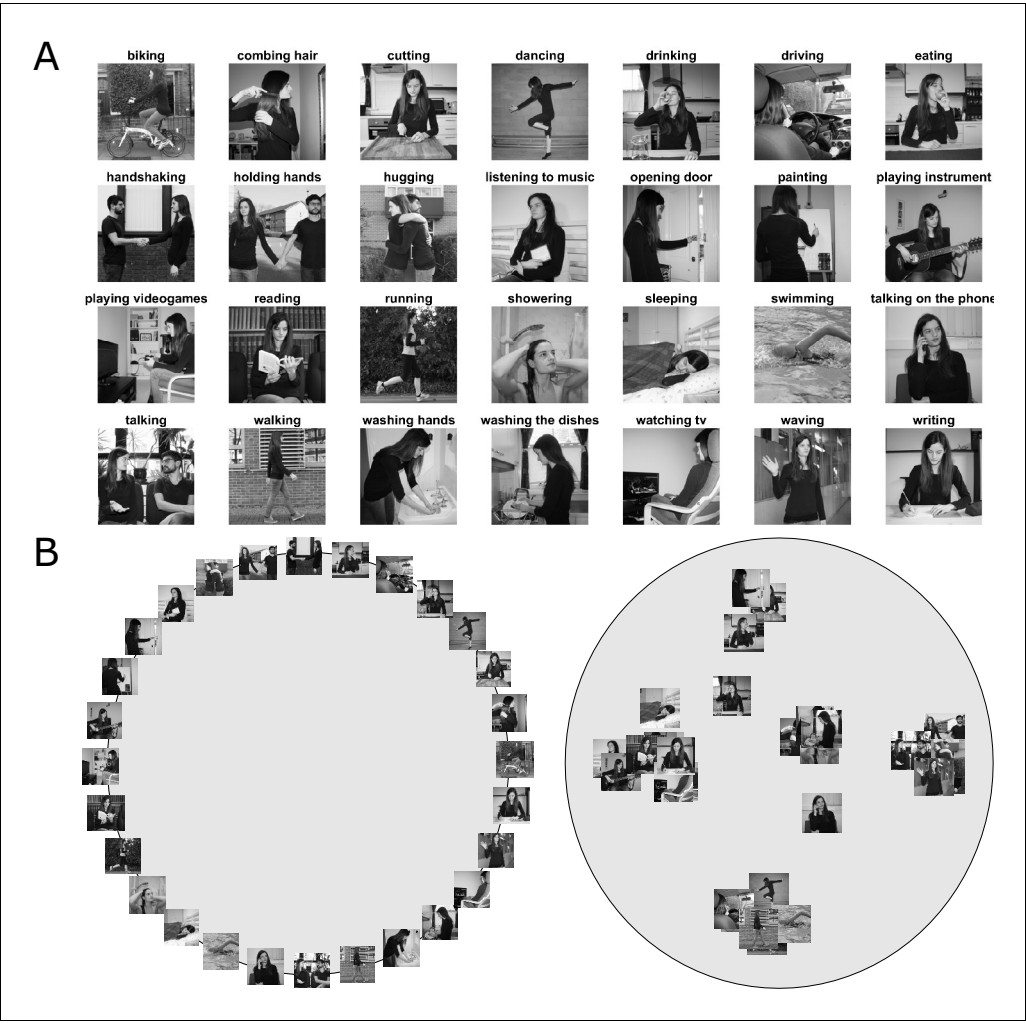

**Figure 1.** Stimuli and behavioral task. (**A**) Stimuli used in the behavioral and the fMRI experiment, depicting static images of 28 everyday actions. To increase perceptual variability, actions were shown from different viewpoints, in different scenes, using two different actors (see text for details) and different objects (for actions involving an object). For a full set of stimuli used in the fMRI experiment, see *Figure 1—figure supplement 1*. (**B**) Illustration of the behavioral experiment used for inverse multidimensional scaling. In the first trial of the experiment, participants were presented with an array of images arranged on the circumference of a gray circle (left panel). In each subsequent trial, an adaptive algorithm determined a subset of actions that provided optimal evidence for the pairwise dissimilarity estimates (see *Kriegeskorte and Mur, 2012* and *Materials and methods*, for details). In different parts of the experiment, participants were asked to rearrange the images according to their perceived similarity with respect to a specific aspect of the action, namely, their meaning (or semantics), the body part(s) involved, the scene/context in which the action typically takes place, movement kinematics, and objects involved in the action. Right panel: Example arrangement resulting from the semantic task of one representative participant. Using inverse multidimensional scaling, we derived a behavioral model (see *Figure 2*) from this arrangement, individually for each participant, that we then used for the representational similarity analysis to individuate those brain regions that showed a similar representational geometry (for details, see *Materials and methods* section).

The online version of this article includes the following figure supplement(s) for figure 1:

**Figure supplement 1.** All images used in the fMRI experiment.

## Results

### Behavioral

Inverse multidimensional scaling experiment (Behavioral)

To obtain representational dissimilarity matrices (RDMs) for the semantic model, we extracted the pairwise Euclidean distances from the participants' inverse MDS arrangements (see Materials and methods, section *Inverse Multidimensional Scaling*, for details). *Figure 2* shows the resulting RDM (averaged across participants).

We found significant (all p-values were smaller than p<0.0001 and survived false discovery rate correction) inter-observer correlations, that is the individual RDMs significantly correlated with the average RDM of the remaining participants (mean leave-one-subject-out correlation coefficient: 0.61, min individual correlation coefficient: 0.46, max individual correlation coefficient: 0.78), suggesting that the participants' arrangements were reliable and based on comparable principles.

### Principal Component Analysis (PCA) and Clustering analysis: K-means

To better characterize the dimensions along which the actions were organized and the clusters resulting from inverse MDS for the semantic task, we carried out a K-means clustering analysis and a

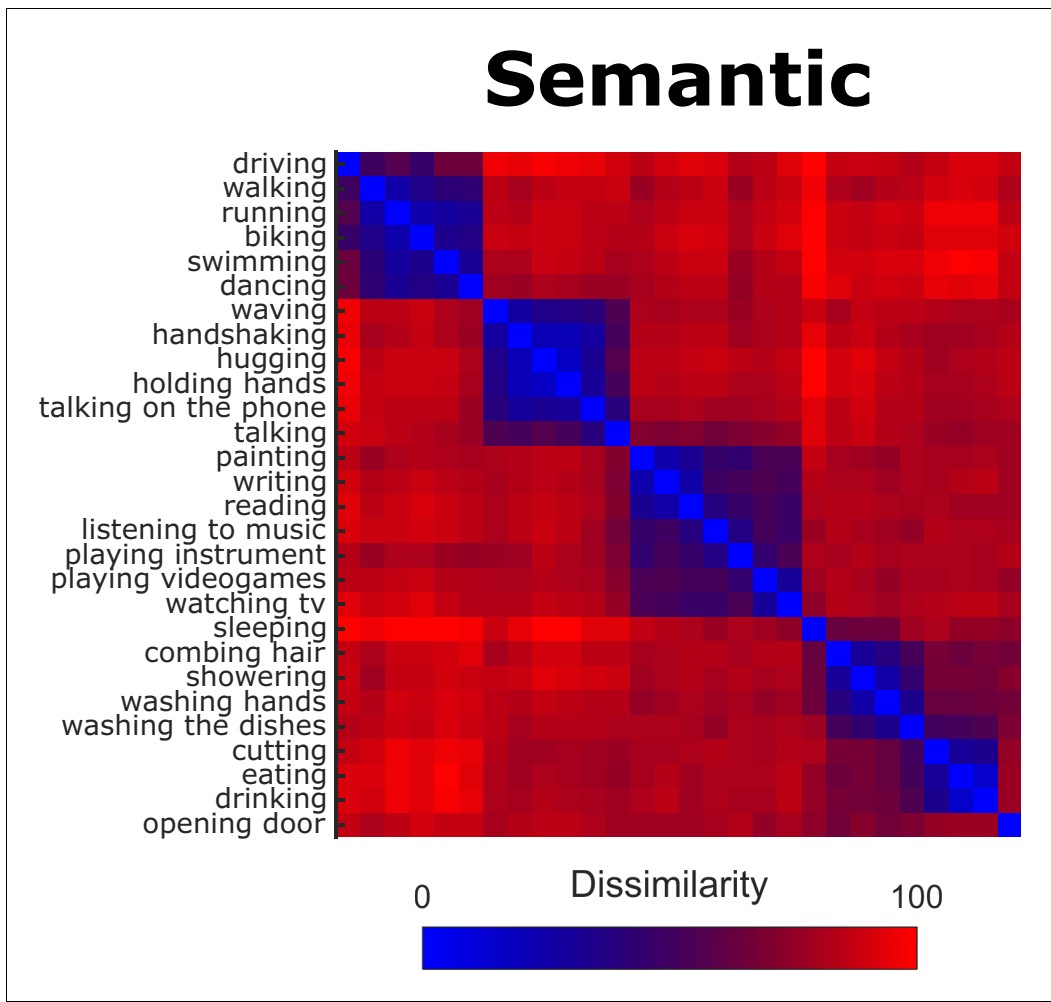

**Figure 2.** Behavioral Representational Dissimilarity Matrix for the semantic model. Bluish colors indicate high similarity between pairwise combinations of actions, whereas reddish colors indicate high dissimilarity.
The online version of this article includes the following figure supplement(s) for figure 2:

**Figure supplement 1.** Pairwise cross-correlation matrix across models.

Principal Component Analysis (PCA; see Materials and methods for details). A Silhouette analysis (see Materials and methods and *Figure 3—figure supplement 1*) revealed that the optimal number of clusters for the semantic task was six. As can be seen in *Figure 3—figure supplement 2*, the first three components account for the largest amount of variance. For ease of visualization, we show the first two components in *Figure 3*. A visualization of the first three principal components is shown in *Figure 3—figure supplement 3*. The analysis revealed clusters related to locomotion (e.g. biking, running), social/communicative actions (e.g. handshaking, talking on the phone), leisure-related actions (e.g. painting, reading), food-related actions (e.g. eating, drinking), and cleaning-related actions (e.g. showering, washing the dishes).

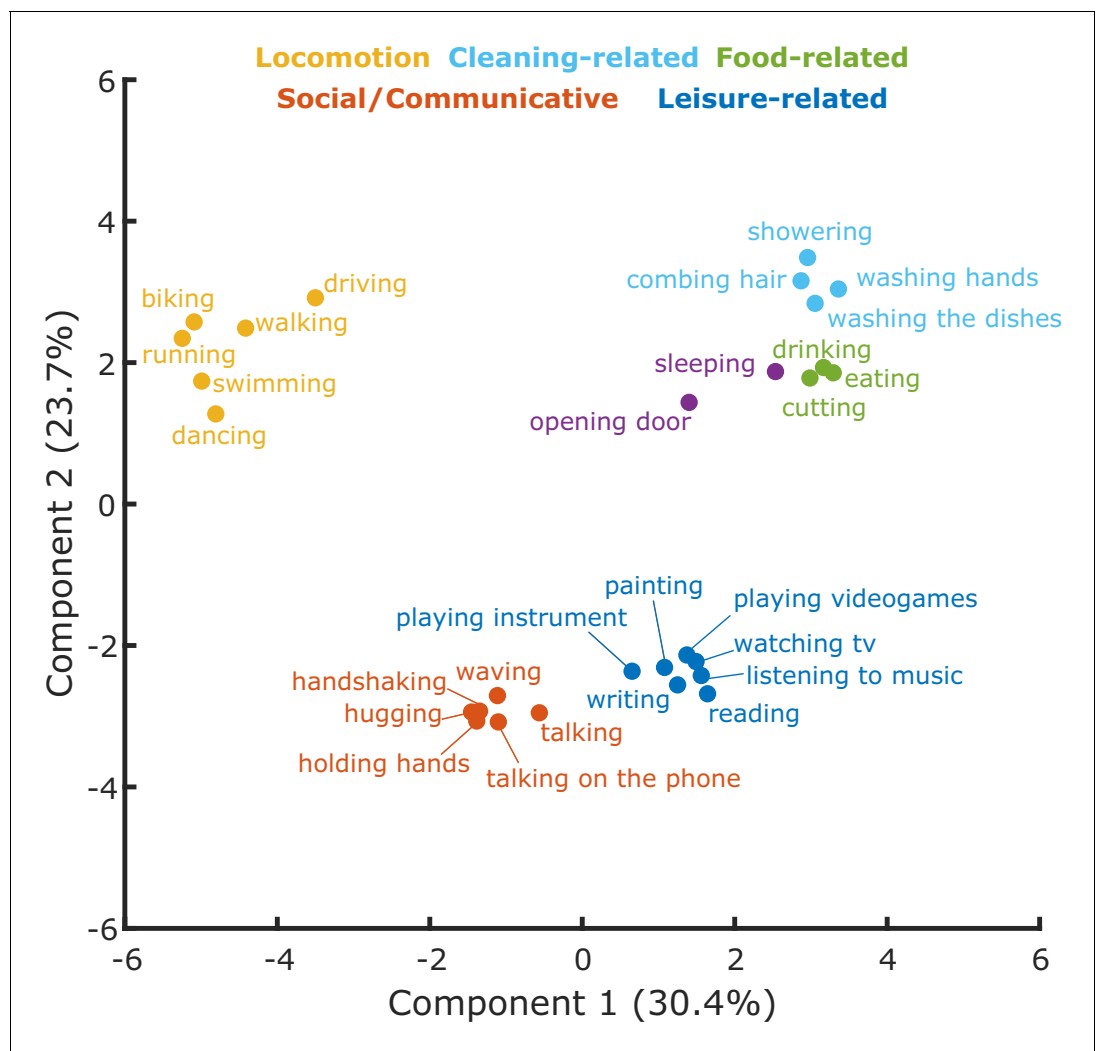

**Figure 3.** Cluster analysis. Clusters resulting from the K-means clustering analysis for the semantic task. The 2D-plot shows component 1 and 2, the corresponding labels of individual actions and the suggested labels for the categories resulting from the K-means clustering. For a visualization of the organization of these actions along the third component, which explained 23.4% of the variance, please refer to *Figure 3—figure supplement 3*. The online version of this article includes the following figure supplement(s) for figure 3:

**Figure supplement 1.** Results silhouette analysis.

**Figure supplement 2.** Eigenvalues obtained from the PCA of the semantic model.

**Figure supplement 3.** Results cluster analysis.

**Figure supplement 4.** First two principal components of the control models.

### RSA

To identify neural representations of observed actions that are organized according to semantic similarity, we performed a searchlight correlation-based RSA using the semantic model derived from the behavioral experiment. We thereby targeted brain regions in which the similarity of activation patterns associated with the observed actions matches the participants' individual behavioral semantic similarity arrangement. We identified significant clusters in bilateral LOTC extending ventrally

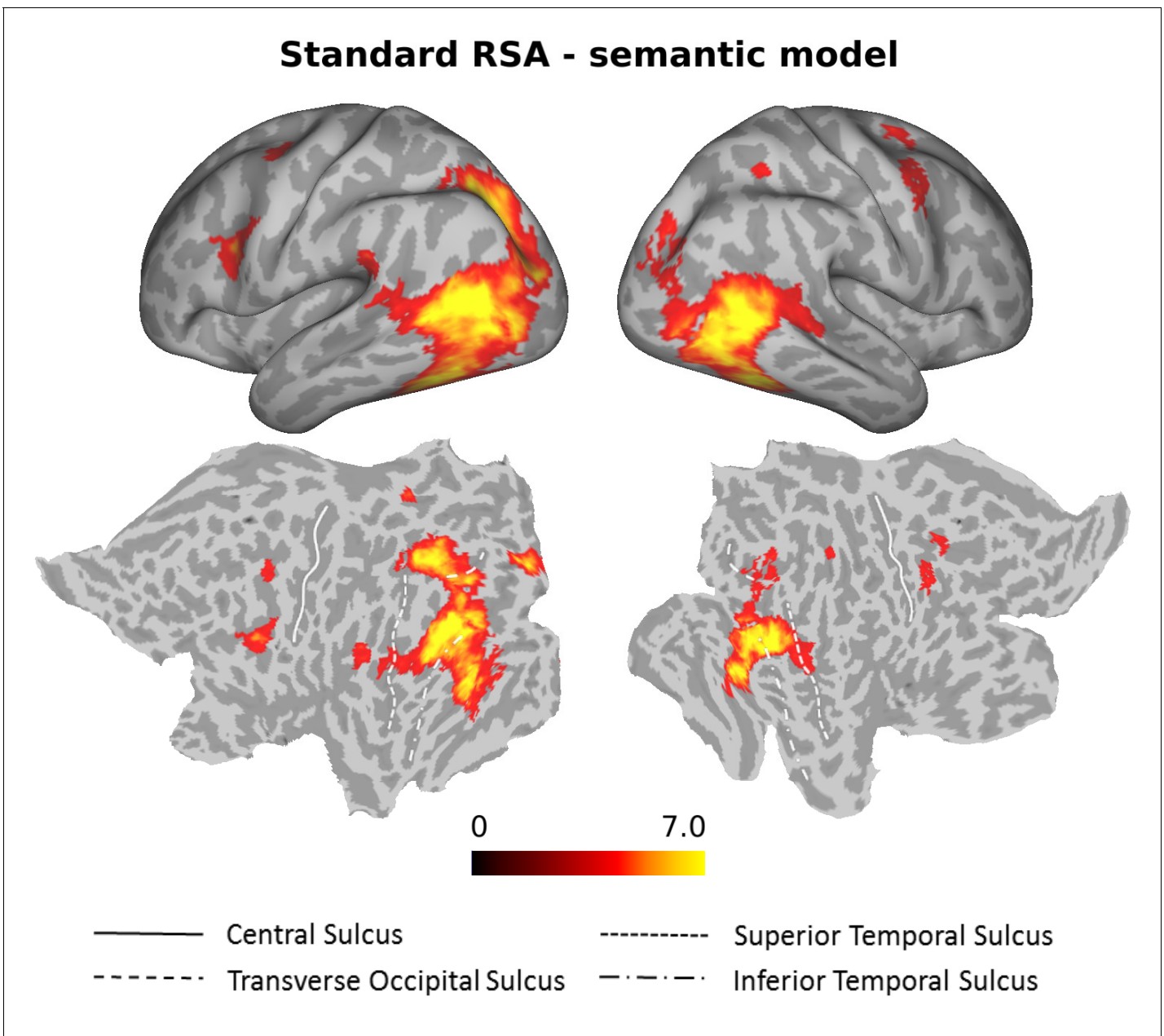

**Figure 4.** Standard RSA, Semantic model. Group results of the searchlight-based RSA using the semantic model (standard RSA, that is second order correlation between neural data and behavioral model). Statistical maps only show the positive t values that survived a cluster-based nonparametric analysis with Monte Carlo permutation (cluster stat: max sum; initial pval <0.001; *Stelzer et al., 2013*). The resulting individual correlation maps were first Fisher transformed and then converted to t scores. After the correction, data were converted to z scores, and only values greater than 1.65 (one-tailed test) were considered as significant. This analysis revealed clusters in bilateral LOTC, bilateral IPL and bilateral precentral gyrus (see *Supplementary file 1* for details).

into inferior temporal cortex, bilateral posterior intraparietal sulcus (pIPS), and bilateral inferior frontal gyrus/ventral premotor cortex (*Figure 4*, *Supplementary file 1*).

Given that a number of action components covary to some extent with semantic features (e.g. locomotion actions typically take place outdoors, cleaning-related actions involve certain objects, etc.; see also *Figure 2—figure supplement 1*), it is impossible to determine precisely what kind of information drove the RSA effects in the identified regions on the basis of the correlation-based RSA alone. Hence, to test which brain areas contain action information in their activity patterns that can be predicted by the semantic model over and above the models for different action components and the three control models, we conducted a multiple regression RSA. We hypothesized that if actions were organized predominantly according to action components (captured in the body, scene, movement, object, sociality and transitivity models) or due to low level differences between conditions (captured in the distance, 1 vs 2 people and HMAX-C1 model; see *Figure 5* and section *Construction of Representational Dissimilarity Matrices* for details), this analysis should not reveal any remaining clusters. Therefore, the multiple regression RSA included ten predictors (semantic, body, scene, movement, object, sociality, transitivity, distance, 1 vs 2 people, HMAX-C1).

As can be seen in *Figure 6*, the semantic model explained significant amounts of variance over and above the six models for the different action components and the three control models in the left anterior LOTC at the junction to the posterior middle temporal gyrus (see also *Figure 6—figure supplement 1* and *Supplementary file 2*).

*Figure 7* shows the results of the multiple regression RSA for the other models (for a visualization of the same results together with the outlines of the Glasser multi-modal parcellation [*Glasser et al., 2016*] superimposed on the flat maps, see *Figure 6—figure supplement 1*). All models except the sociality and the scene model revealed significant clusters (see *Supplementary file 2* for details). The clusters obtained for the body, movement distance, and the 1 vs 2 people models that explained significant amounts of variance over and above the remaining models partially overlapped with the cluster revealed by the semantic model (*Figure 6*; outlines superimposed in black in *Figure 7* for ease of comparison). For the body model (*Figure 7A*), the cluster was obtained predominantly in the left angular gyrus, slightly more dorsal than the cluster revealed by the semantic model, and the right pMTG. Note that the cluster revealed by the body model in the left hemisphere was more dorsal than Extrastriate Body Area (EBA) (*Downing et al., 2001*) (for a comparison with coordinates revealed by previous studies that used a functional localizer to identify EBA, see *Figure 6—figure supplement 2*). The movement model revealed clusters in the left SPL, the left inferior occipital gyrus (posterior to the cluster revealed by the semantic model) and the right MTG (*Figure 7B*). The object model explained variance in the right intraparietal sulcus area 1 (IPS1; *Hagler et al., 2007*) and the left fusiform gyrus area (*Figure 7C*). The transitivity model revealed a small cluster in the left inferior occipital gyrus, posterior to the cluster revealed by the semantic cluster (*Figure 7D*). The distance (*Figure 7E*) and the 1 vs 2 people model (*Figure 7F*) revealed a number of clusters, mostly posterior to the cluster revealed by the semantic model, both in the left and right hemisphere (for a full set of labels and peak coordinates, see *Supplementary file 2*). The HMAX-C1 model revealed clusters (distinct from the cluster revealed by the semantic model) in the calcarine cortex (left hemisphere), the right occipital fusiform gyrus, the right inferior occipital gyrus, the right lingual gyrus and the left fusiform gyrus (*Figure 7G* and *Supplementary file 2*).

Note that the obtained RSA results are unlikely to be due to some low-level features of the images we used. First, to minimize the risk that results could be driven by trivial pixel-wise perceptual similarities, we introduced a substantial amount of variability for each of the 28 actions, using different exemplars in which we varied the actor, the viewpoint, the background/scene (e.g. kitchen A vs kitchen B), and the object (for actions that involved one). Second, as described above, to account for low-level perceptual similarities, we included three control models (distance, one vs two people, HMAX-C1) in the multiple regression analysis. The resulting clusters that show the variance explained by these control models can be seen in *Figure 7 (E–G)* and *Figure 6—figure supplement 1*.

To have a better idea of the representational geometry encoded in the left LOTC cluster, we extracted the beta estimates associated with the 100 features neighboring the vertex with the highest T score in the cluster in the left LOTC revealed by the multiple regression RSA for the semantic model. For this visualization, we used exactly the same steps involved in the searchlight-based standard RSA (see *Materials and methods* section for details). We derived the RDM from the beta

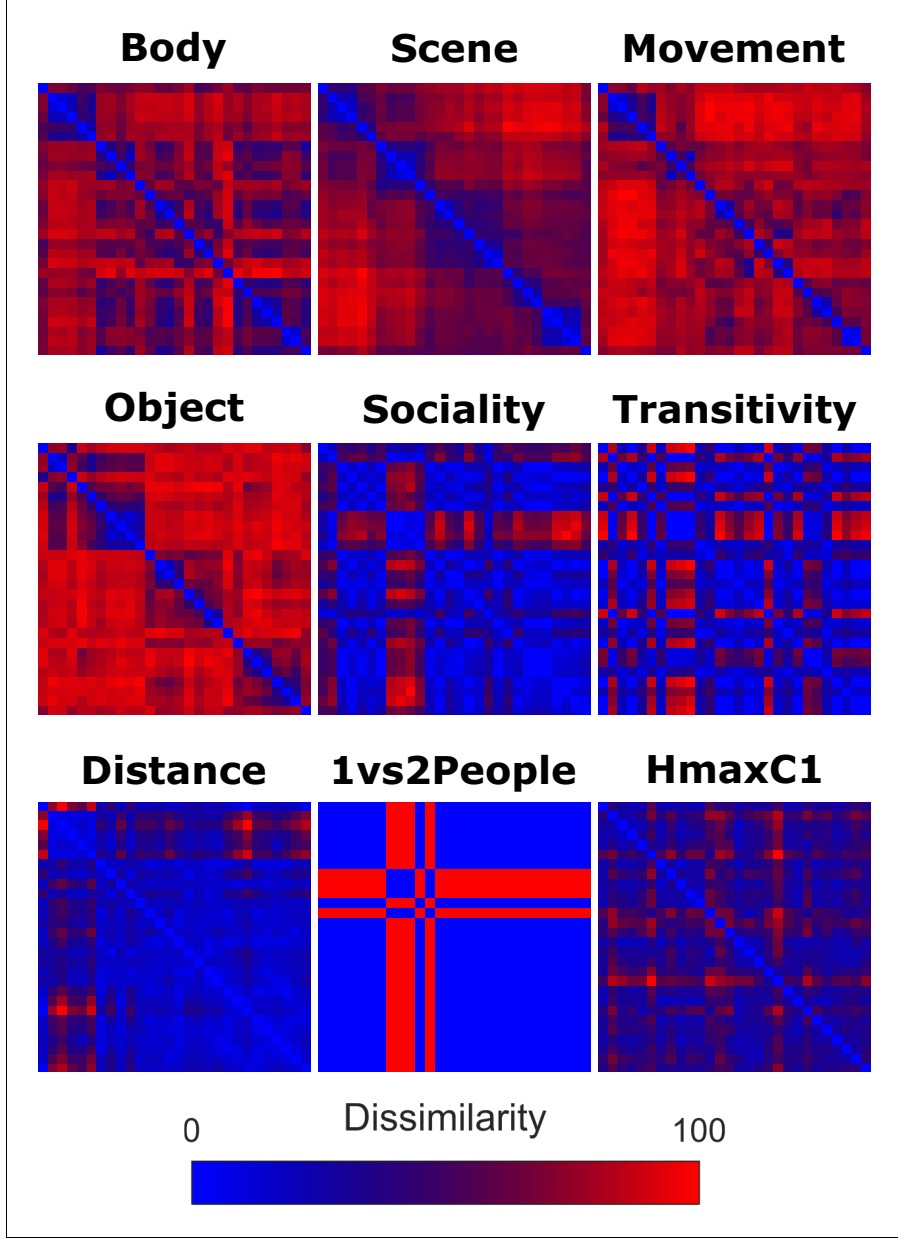

**Figure 5.** Model Representational Dissimilarity Matrices. Group representational dissimilarity matrices for body, scene, movement, and object model, derived from inverse MDS carried out in the same group of participants after the fMRI experiment. The sociality, transitivity and distance model were based on ratings in a separate group of participants. For construction of the 1 vs 2 people and the HMAX-C1 model, and further details on the construction of the remaining models, see Materials and methods, section Construction of Representational Dissimilarity Matrices. Bluish colors indicate high similarity between pairwise combinations of actions, whereas reddish colors indicate high dissimilarity. For ease of comparison, we used the ordering of actions resulting from the semantic task (see **Figure 2**).

patterns extracted from the ROI and then correlated this neural RDM with the behavioral model RDMs (semantic, body, scene, movement, object, sociality, transitivity) and the control model RDMs (distance, 1 vs 2 People, HMAX-C1). The averaged correlations between the model RDMs and the neural RDM in the LOTC obtained from this analysis are reported in *Figure 8A*. The bar plot confirmed that the semantic model is the model that best correlates with the neural RDM (shown in *Figure 8B*). Not surprisingly, also the other models significantly correlated with the neural RDMs, with the exception of the scene, sociality, object and HMAX-C1 models. Note that the averaged

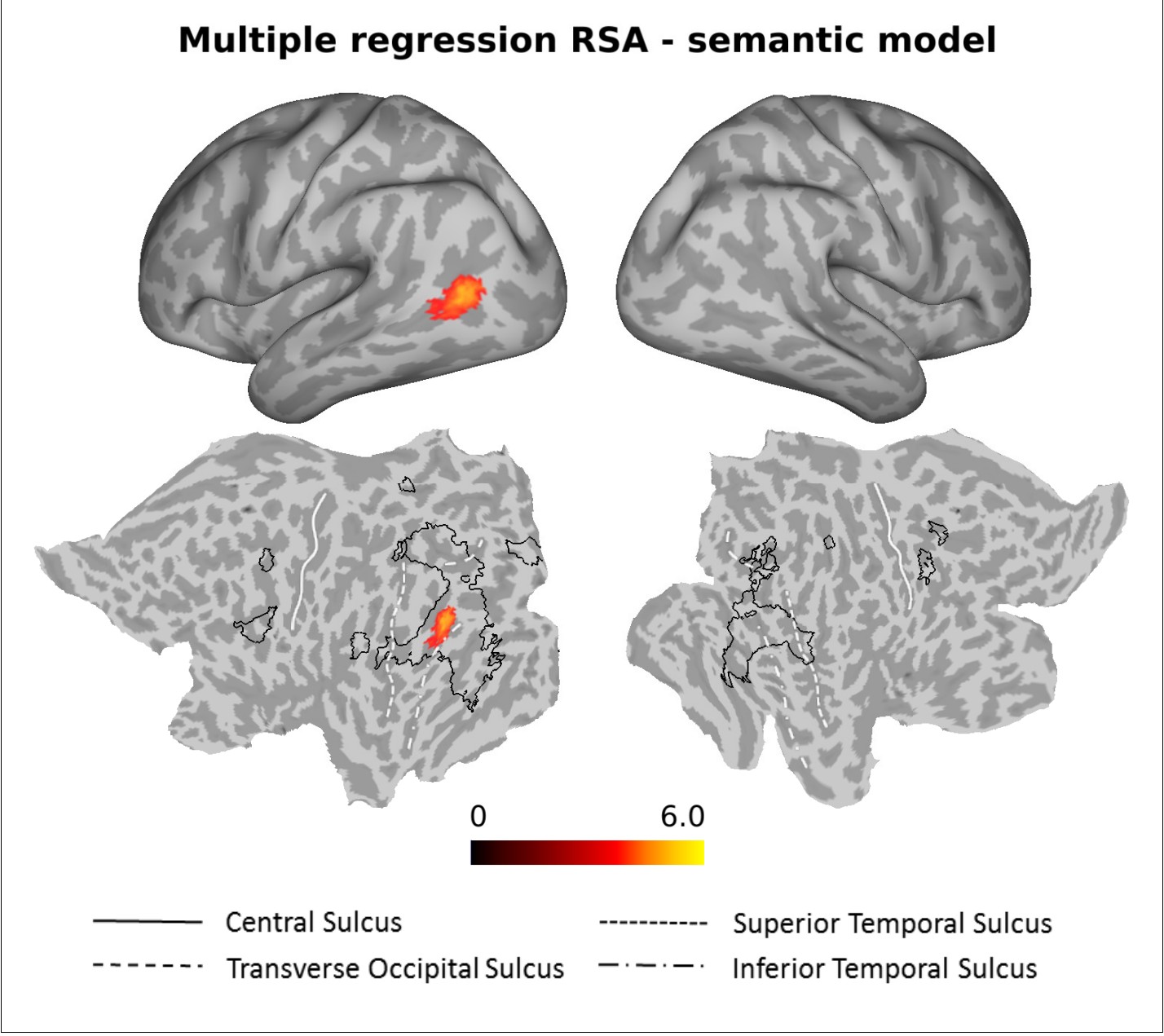

**Figure 6.** Multiple regression RSA (semantic model). Group results of the searchlight-based multiple regression RSA, in which the ten different models (see *Materials and methods*, section *Construction of Representational Dissimilarity Matrices*, for details) were used as regressors in a multiple regression RSA conducted at the individual level. The resulting beta estimates were converted to t scores across participants and then corrected for multiple comparisons using cluster-based nonparametric permutation analysis (*Stelzer et al., 2013*) (see *Materials and methods* for details). Accounting for the behavioral (body, scene, movement, object, sociality, transitivity) and the control models (distance, 1 vs 2 People, HMAX-C1) in the multiple regression analysis, the semantic model explained observed variance in the left LOTC. For ease of comparison, the black outlines in the bottom part of the figure (flat maps) depict the clusters revealed by the standard RSA for the semantic model (*Figure 4*).

The online version of this article includes the following figure supplement(s) for figure 6:

**Figure supplement 1.** Multiple regression based RSA results, together with Glasser parcellation.

**Figure supplement 2.** Results for the body model, together with EBA coordinates.

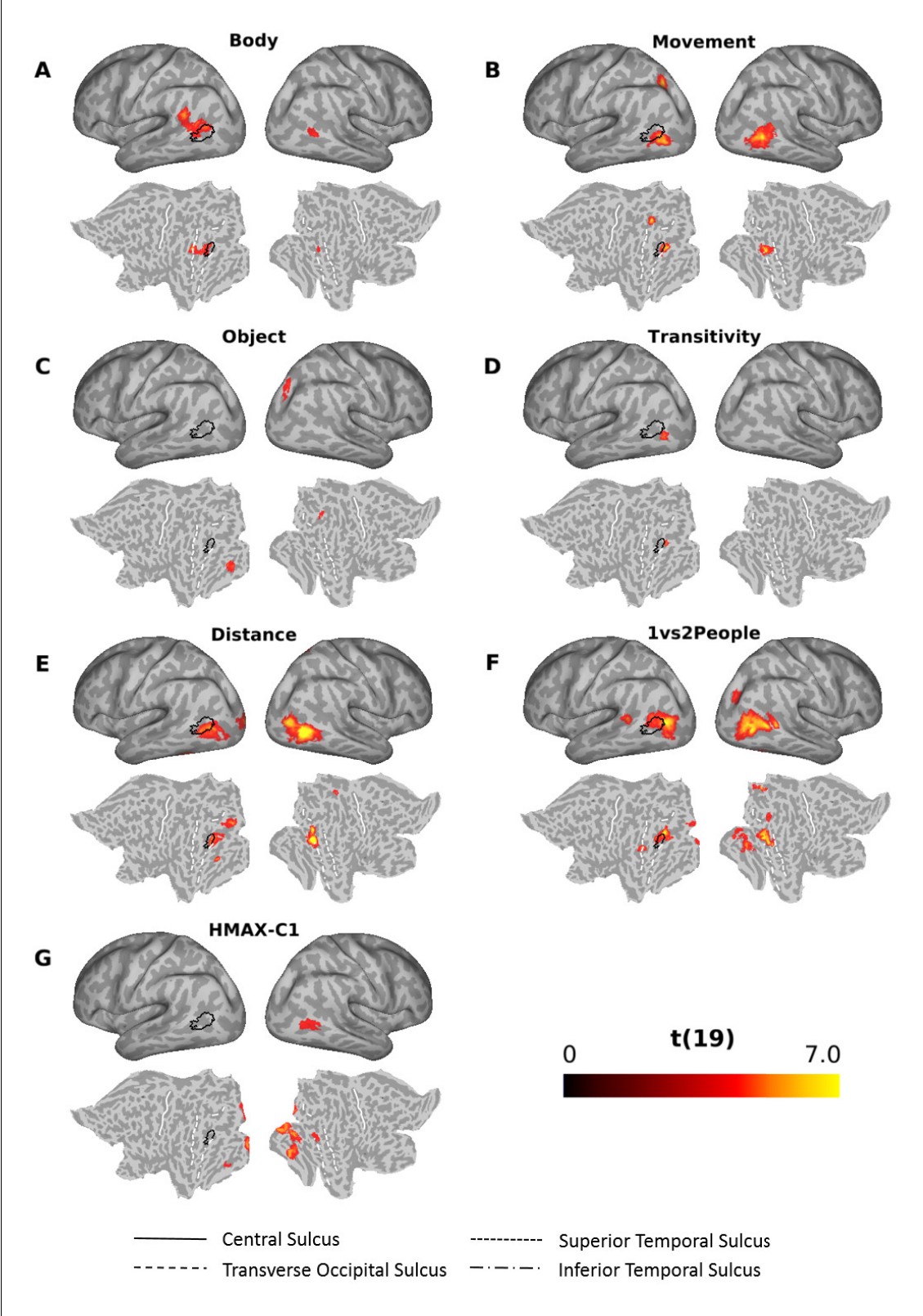

**Figure 7.** Searchlight-based multiple regression RSA. Searchlight-based multiple regression RSA results for the body (**A**), movement (**B**), object (**C**), transitivity (**D**), distance (**E**), 1 vs 2 people (**F**) and the HMAX-C1 (**G**) model. The resulting beta estimates were converted to t scores across participants and then corrected for multiple comparisons using cluster-based nonparametric permutation analysis (*Stelzer et al., 2013*) (see *Materials and methods*
*Figure 7 continued on next page*

*Figure 7 continued*

for details). Results for the scene and the sociality model did not survive corrections for multiple corrections and thus are not shown here. Black outlines on the inflated brains and the flat maps depict significant clusters revealed by the multiple regression RSA for the semantic mode (*Figure 6*).

correlations were quite low (semantic model: 0.0645, distance model: 0.0608, 1vs2People: 0.0563; all other models < 0.05), in line with a number of previous studies (e.g. *Bracci and Op de Beeck, 2016*; *Magri et al., 2019*). To visualize how the data are organized in a two-dimensional space and thus to better understand the underlying representational geometry encoded in the LOTC, we conducted a classical multidimensional scaling (MDS) analysis on the neural RDMs averaged across participants. As shown in *Figure 8C*, the organization of the neural multidimensional patters associated

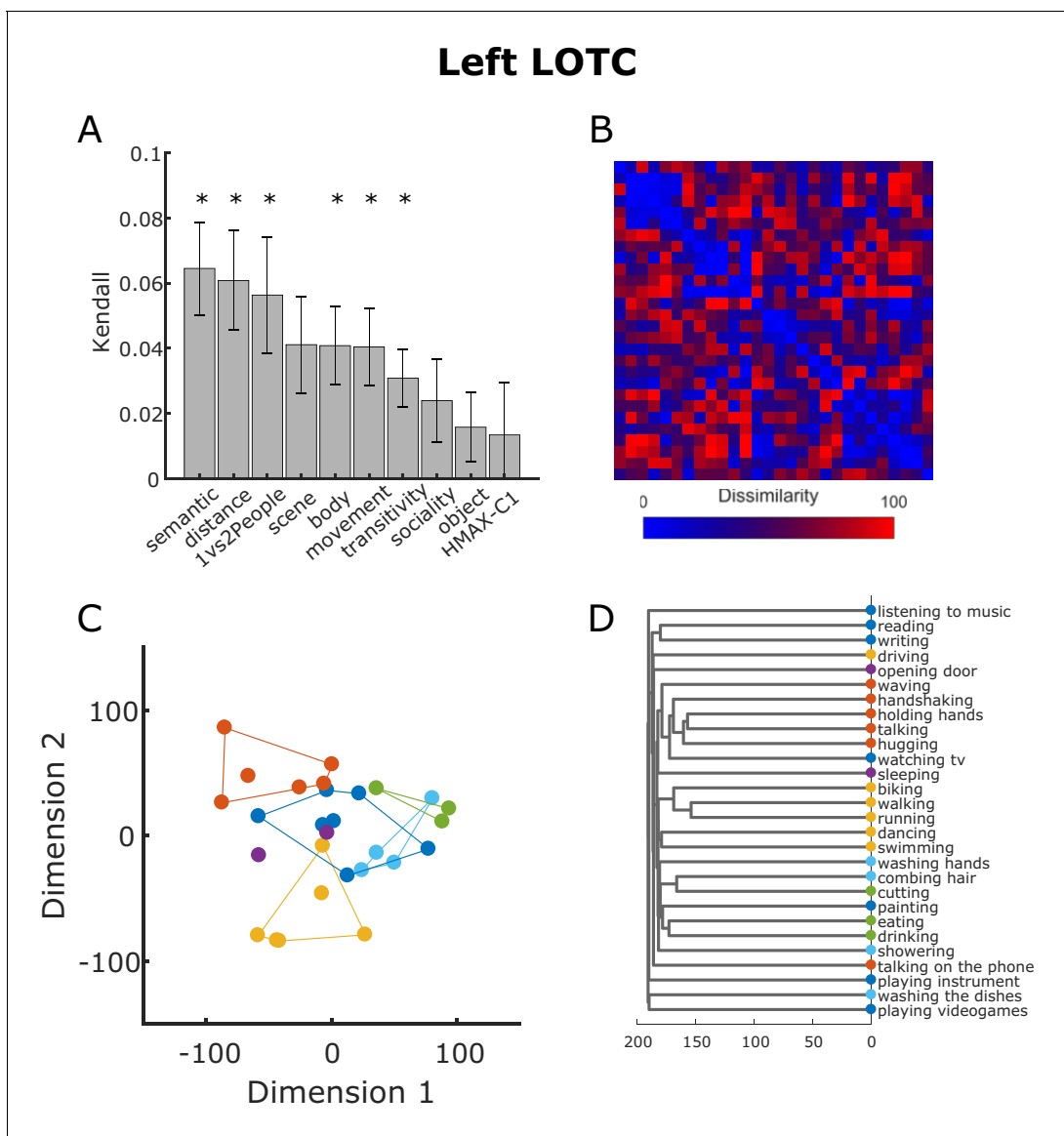

**Figure 8.** Neural RDM from LOTC. (A) Kendall rank correlation coefficient between neural and model RDMs in the cluster in the LOTC revealed by the multiple regression RSA using the semantic model (see Figure 6). Error bars depict the standard error of the mean. Asterisks indicate that the correlation was greater than zero (one-tailed t-test, FDR corrected). (B) Neural RDM from the LOTC used for the analysis shown in panel (A). Values were separately rank-transformed and then rescaled to values between 0 and 100, as suggested in *Nili et al. (2014)*. (C) Two-dimensional visualization of the beta patterns of the cluster in the LOTC resulting from classical MDS. Actions were assigned to the clusters resulting from the behavioral analysis as shown in *Figure 3*. (D). Dendrogram resulting from the hierarchical clustering analysis.

with each action indeed resembles the clustering organization we observed for the behavioral semantic task (see *Figure 3*). To facilitate the comparison with *Figure 3*, we assigned actions with the same color code as the corresponding clusters identified from the behavioral analysis. A hierarchical clustering analysis showed a similar result (see *Figure 8D*).

## Discussion

Here we aimed to investigate the organizational principles of everyday actions and the corresponding neural representations. Using inverse MDS (*Kriegeskorte and Mur, 2012*), we identified a number of clusters emerging in the arrangement of actions according to their meaning that were relatively stable across participants. These clusters corresponded to meaningful categories, namely locomotion, communicative actions, food-related actions, cleaning-related actions, and leisure-related actions (*Figure 3*). Using multiple regression RSA (*Kriegeskorte et al., 2008a*), we showed that this semantic similarity structure of observed actions was best captured in the patterns of activation in the left LOTC.

PCA suggested that the five categories revealed in the behavioral experiment appear to be organized along three main components that together explained around 78% of the observed variance. Whereas neither the PCA nor the k-means clustering provides objective labels of the main dimensions that might underlie the organization of actions into categories, it appears that clusters corresponding to semantic categories along the first principal component differed with respect to the type of change induced by the action (negative side of component 1: change of location, positive side of component 1: change of external/physical state, middle: change of internal/mental state). The second component seemed to distinguish actions that fulfil basic (or physiological) needs such as eating, drinking, cutting, or getting from one place to the other, and actions that fulfil higher (social belonging, self-fulfillment) needs such as hugging, talking to someone, reading, listening to music). Interestingly, this distinction shows some similarity with Maslow's hierarchy structure of needs (*Maslow, 1943*). The third component might capture the degree to which an action is directed towards another person (hugging, holding hands, talking, etc.) or not (running, swimming, playing video games, reading). To the best of our knowledge, the only study that explicitly examined behavioral dimensions underlying the organization of actions focused on tools and the way they are typically used. Using ratings of the use of tools depicted as static images, *Watson and Buxbaum (2014)* found that the two components that best explained the variability of their ratings of tool-related actions were related to 'hand configuration' and 'magnitude of arm movement'.

In the language domain, there exists a rich literature on the semantic structure of action representations. As an example, based on the examination of the relationship between verb meaning (e.g. to break, to appear) and verb behavior (e.g. whether or not a verb can be used transitively), a number of authors including *Talmy (1985)*, *Levin (1993)*, and *Pinker (1989)* aimed to reveal the underlying semantic structure of verbs (for related work on the use of semantic feature production norms see for example *Vinson and Vigliocco, 2008*). Based on their analyses, these authors proposed a number of *semantic categories* (*Talmy, 1985*) or *semantic fields* (*Levin, 1993*). The latter include, but are not limited to, *change of location, communication, cooking*, and *change of state*, which show similarities to the action categories *locomotion, communicative actions, food-related actions* and *cleaning-related actions* revealed in the current study. It is less clear how to map the category we labeled *leisure-related actions* onto semantic fields proposed by *Levin (1993)* and others. Whereas it is not surprising that action categories revealed on the basis of visual stimuli to some degree resemble semantic fields derived from cross-linguistic comparisons, it is likely that there exist action categories that can be revealed by visual stimuli but not by language, and vice versa (see also *Vinson and Vigliocco, 2008*, and *Watson and Buxbaum, 2014*, for related discussions on this topic). As an example, actions depicted by visual stimuli are concrete depictions, whereas actions described by words by definition are symbolic and thus more abstract representations of actions. Based on the existing literature on semantic categories, we expect that future studies, using a similar approach as described in the current study with a wider range of actions, will reveal additional categories and overarching dimensions that show similarities, but are not necessarily identical to the semantic categories described in the literature so far.

To identify brain regions that encoded the similarity patterns revealed by the behavioral experiment, we conducted a searchlight-based RSA. We observed a significant correlation between the

semantic model and the pattern of fMRI data in regions of the so-called action observation network (*Caspers et al., 2010*), which broadly includes the LOTC, IPL, premotor and inferior frontal cortex. Using multiple regression RSA, we found that only the left LOTC contains action information as predicted by the semantic model over and above the remaining models. In line with these results, it has been demonstrated that it is possible to discriminate between observed actions based on the patterns of activation in the LOTC, generalizing across objects and kinematics (*Wurm et al., 2016*), and at different levels of abstraction (*Wurm and Lingnau, 2015*). Interestingly, studies using semantic tasks on actions using verbal stimuli (*Watson et al., 2013*) or action classification across videos and written sentences (*Wurm and Caramazza, 2019a*) tend to recruit anterior portions of the left LOTC. By contrast, studies using static pictures (*Hafri et al., 2017*) or videos depicting actions (*Hafri et al., 2017*; *Wurm et al., 2017*; *Wurm et al., 2016*; *Wurm and Lingnau, 2015*) find LOTC bilaterally and more posteriorly, closer to the cluster in the MOG identified in the current study. Note that a previous study by *Hafri et al. (2017)*, directly comparing areas in which it is possible to decode between actions depicted by static pictures and by dynamic videos, found a far more widespread network of areas decoding actions depicted by videos in comparison to static pictures. We thus cannot rule out that more dynamic action components, for example movement kinematics, are not well captured in the current study. At the same time, *Hafri et al. (2017)* demonstrated that it is possible to decode actions across stimulus format in the posterior LOTC, in line with the view that action-related representations in this area does not necessarily require visual motion. Together, our findings suggest that this area captures semantic aspects of actions at higher-order visual levels, whereas anterior portions of the left LOTC might capture these aspects at stimulus-independent or verbal levels of representation (see also *Lingnau and Downing, 2015* and *Papeo et al., 2019*).

Whereas the focus of the multiple regression RSA was to examine the results for the semantic model while accounting for the variability explained by the remaining models, it is interesting to compare the clusters revealed by the remaining models and how they relate to the semantic model. Several models (body, movement, distance, 1 vs 2 people) revealed clusters that were partially distinct from and partially overlapped with the cluster revealed by the semantic model. By contrast, the transitivity model revealed a cluster in the ventral portion of the LOTC that did not overlap with the semantic model. The LOTC has been shown to be sensitive to categorical action distinctions, such as whether they are directed towards persons or objects (*Wurm et al., 2017*). The results obtained for the transitivity model are in line with the results reported by *Wurm et al. (2017)* and *Wurm and Caramazza (2019b)*, whereas we failed to obtain reliable results for the sociality model. Note, however, that the current study was not designed to test this model. In particular, only a small number of actions were directed towards another person, and these actions co-varied with the presence of another person, which was captured by the 1 vs 2 people model (which showed comparable results to those obtained for the sociality model in the study by *Wurm et al., 2017*; see also *Wurm and Caramazza, 2019b* for an experimental segregation of sociality and 1 vs. 2 people models).

Together, the results of the multiple regression RSA obtained for the different models are in line with the view that the LOTC hosts a variety of different, partially overlapping representations of action components that are likely to be integrated flexibly according to task demands (see also *Lingnau and Downing, 2015*). Multiple, possibly overlapping basic dimensions have been proposed to underlie the organization of these different action components within the LOTC, among them the input modality (visual versus non-visual; *Lingnau and Downing, 2015*; *Papeo et al., 2019*), the presence and orientation of a person, and the directedness of actions toward persons or other targets (*Wurm et al., 2017*; *Wurm and Caramazza, 2019b*).

Whereas the standard RSA revealed strong clusters not only in the LOTC but also in the IPL and the precentral gyrus, the multiple regression RSA for the semantic model revealed a cluster in the left LOTC only. This observation suggests that the results obtained in the IPL and the precentral gyrus for the semantic model revealed by the standard RSA was due to some combination of the models accounted for in the multiple regression RSA, even if no individual model alone revealed a cluster in these two regions that survived corrections for multiple comparisons.

Our results call for a comparison with the object domain, where similar questions have been addressed for decades. In line with the results of the multiple regression-based RSA, the LOTC has been demonstrated to represent the similarity structure of object categories (*Bracci and Op de Beeck, 2016*). Regarding the results of the cluster analysis, salient distinctions at the behavioral and neural level have been found between animate and inanimate objects (for example

*Kriegeskorte et al., 2008b*; *Chao et al., 1999*; *Caramazza and Shelton, 1998*) which have been further segregated into human and nonhuman objects (*Mur et al., 2013*), and manipulable and non-manipulable objects (*Mecklinger et al., 2002*), respectively. The division between animate and inanimate objects, supported by neuropsychological, behavioral and neuroimaging findings, has been suggested to have a special status, likely due to evolutionary pressures that favored fast and accurate recognition of animals (*Caramazza and Shelton, 1998*; *Mur et al., 2013*; *New et al., 2007*). We conjecture that similar evolutionary mechanisms might have produced the distinction between actions belonging to different categories, such as locomotion (which might indicate the approach of an enemy), food-related actions (which might be critical for survival) and communicative actions (critical for survival within a group).

## Conclusions

Using a combination of behavioral and fMRI data, we identified a number of meaningful semantic categories according to which participants arrange observed actions. The corresponding similarity structure was captured in left LOTC over and above the major components of perceived actions (body parts, scenes, movements, and objects) and other related features of the observed action scenes, in line with the view that the LOTC hosts a variety of different, partially overlapping action components that can be integrated flexibly. Together, our results support the view that the LOTC plays a critical role in accessing the meaning of actions beyond the mere perceptual processing of action-relevant components.

# Materials and methods

## Participants

Twenty healthy participants (13 females; mean age: 28 years; age range: 20–46) took part in an fMRI and a behavioral experiment carried out at the Combined Universities Brain Imaging Centre (CUBIC) at Royal Holloway University of London (RHUL). The experiment was approved by the ethics committee at the Department of Psychology, RHUL (REF 2015/088). Participants provided written informed consent before starting the experiment. All participants were right-handed with normal or corrected-to-normal vision and no history of neurological or psychiatric disease. All participants but one (RT, one of the authors) were naïve to the purpose of the study.

## Inverse multidimensional scaling

Participants sat in front of a monitor (LCD 16.2 × 19.2 inches; distance 60 cm). In trial 1, all action images (one exemplar per action) appeared on the screen in a circular arrangement (with the order of actions randomly selected; see *Figure 1B*). Participants were instructed to arrange the pictures by drag-and-drop using the mouse according to their perceived similarity in meaning (e.g. *walking* and *running* would be placed closer to each other than *walking* and *drinking*) and to press a button once they were satisfied with the arrangement. In each subsequent trial (trial two to $N_p$, where $N_p$ is the total number of trials for participant *p*), a subset of stimuli was sampled from the original stimulus set. The subset of actions was defined using an adaptive algorithm that provided the optimal evidence for the pairwise dissimilarity estimates (which are inferred from the 2D arrangement of the items on the screen, see *Kriegeskorte and Mur, 2012* for details). Participants were given 15 minutes in total to complete the task.

## Stimulus selection

In contrast to previous studies that used a small set of actions, we aimed to cover a wide range of actions that we encounter on a daily basis. To this aim, we initially carried out an online survey using Google Forms. The aim of the survey was to identify actions that are considered common by a large sample of people. We thus asked 36 participants (different from those that took part in the fMRI study) to spontaneously write down all the actions that came to their mind within 10 min that they or other people are likely to do or observe. As expected, participants often used different words to refer to similar meanings (e.g. talking and discussing) or used different specific objects associated with the same action (e.g. drinking coffee and drinking water). Two of the authors (EB, AL) thus assigned these different instances of similar actions to a unique label. Actions were selected if they

were mentioned by at least 20% of the participants. In total, we identified 37 actions (see *Supplementary file 3*).

As a next step, we selected a subset of actions that were best suitable for the fMRI experiment. Specifically, we aimed to choose a set of actions that were arranged consistently across participants according to their perceived similarities in meaning. To this aim, we retrieved images depicting the 37 actions from the Internet. Using these images, we carried out inverse MDS (see corresponding section and *Kriegeskorte and Mur, 2012* for details) using 15 new participants. Each participant had 20 min to complete the arrangement. In three additional 20 min sessions, participants were furthermore instructed to arrange the actions according to the perceived similarity in terms of their meaning, the scenes in which these actions typically take place, movement kinematics, and the objects which are typically associated with these actions. The order in which these four tasks (semantics, scenes, movements, objects) were administered to participants was counterbalanced across participants. To rule out that any obtained arrangements were driven by the specific exemplars chosen for each action, we repeated the same experiment with a new group of people (N = 15) and an independent set of 37 images taken from the Internet.

To construct representational dissimilarity matrices (RDMs), we averaged the dissimilarity estimates for each pair of actions (e.g. the dissimilarity between biking and brushing teeth, etc.), separately for each participant and each task, across trials. For each participant and model, we then constructed dissimilarity matrices based on the Euclidean distance between each pair of actions that resulted from the inverse MDS experiment. The dissimilarity matrices were then normalized by dividing each value by the maximum value of each matrix. Each row of this matrix represented the dissimilarity judgment of one action with respect to every other action. To select the most suitable actions for the fMRI experiment, we aimed to evaluate which of the 37 actions were arranged similarly across participants in the different tasks. To this aim, we carried out a cosine distance analysis, which allowed us to determine, for each action, the similarity across all participants. The cosine distance evaluates the similarity of orientation between two vectors. It can be defined as one minus the cosine of the angle between two vectors of an inner product space: a cosine distance of 1 indicates that the two vectors are orthogonal to each other (maximum dissimilarity/minimum similarity); a cosine distance of zero indicates that the two vectors have the same orientation (maximum similarity/minimum dissimilarity). The cosine distance can therefore range between 0 and 1. In an RDM, each row (or column) represents the dissimilarity score between one action and every other action, ranging from 0 (minimum dissimilarity) to 1 (maximum dissimilarity). Therefore, each row of the matrix of each single participant was used to compute the pairwise cosine distances between this and the corresponding row of every other participant. For each action, a cosine distance close to zero would indicate that participants agreed on the geometrical configuration of that action with respect to every other action; a value close to one would indicate disagreement. For each action, we computed the mean across the pairwise cosine distances of all participants in both behavioral pilot experiments and kept only those actions that had a cosine distance within one standard deviation from the averaged cosine distance in all tasks and both stimulus sets. Thirty-one actions fulfilled this criterion, whereas five (*getting dressed*, *cleaning floor*, *brushing teeth*, *singing* and *watering plants*) had to be discarded. We also decided to remove two additional actions (*grocery shopping* and *taking the train*) because these could not be considered as single actions but implied a sequence of actions (e.g. *entering the shop, choosing between products, etc.; waiting for the train, getting on the train, sitting on the train, etc.*).

At the end of the procedure, we identified 30 actions that could be used for the next step, which consisted in creating the final stimulus dataset. To this aim, we took photos of 29 of the 30 actions using a Canon EOS 400D camera. To maximize perceptual variability within each action, and thus to minimize low-level feature differences between actions, we varied the actors (2), the scene (2) and perspectives (3), for a total of 12 exemplars per action. Exemplars for the action 'swimming' were collected from the Internet because of the difficulties in taking photos in a public swimming pool.

The distance between the camera and the actor was kept constant within each action (across exemplars). Since some actions consisted of hand-object interactions (such as painting, drinking) and thus required finer details, while other actions involved the whole body (such as dancing, running) and thus required a certain minimum distance to be depicted, it was not possible to maintain the same distance across all the actions. The two actors were instructed to maintain a neutral facial expression and were always dressed in similar neutral clothing. If an action involved an object, the

actor used two different exemplars of the object (e.g. two different bikes for *biking*) or two different objects (e.g. a sandwich or an apple for *eating*). Furthermore, some actions required the presence of an additional actor (*handshaking, hugging, talking*). The brightness of all pictures was adjusted using PhotoPad Image Editor (www.nchsoftware.com/photoeditor/). Pictures were then converted into grayscale and resized to 400 × 300 pixels using FastOne Photo (www.faststone.org). In addition, we made the images equally bright using custom written Matlab code available at: osf.io/cvrb2 (*Tucciarelli, 2019*) (mean brightness across all images was 115.80 with standard deviation equal to 0.4723).

To ensure that the final set of pictures were comprehensible and identified as the actions we intended to investigate, we furthermore validated the stimuli through an online survey using Qualtrics and Amazon Mechanical Turk involving 30 participants. Specifically, the 30*12 = 360 pictures were randomly assigned to three groups of 120 images. Each group was assigned to ten participants that had to name the actions depicted in the images. For each participant, the images were presented in a random sequence. Since most of the participants failed to correctly name some of the exemplars of *making coffee* and *switching on lights*, these actions were excluded from the stimulus set. Therefore, the final number of actions chosen for the main experiment was 28 (see *Supplementary file 3*).

Note that we decided to use static images instead of videos of actions for two reasons. First of all, we wished to avoid systematic differences between conditions based on the kinematic profiles of videos of actions. Second, we aimed to use stimuli that are suitable both for the fMRI experiment and for the inverse MDS experiment. We considered static stimuli more suitable for the latter, given that the participant had to judge the similarity of a large set of actions simultaneously. We examine the consequences of this choice in the discussion.

## Experimental design and setup

The fMRI experiment consisted of twelve functional runs and one anatomical sequence halfway through the experiment. Each functional run started and ended with 15 s of fixation. In between runs, the participants could rest.

Stimuli were back-projected onto a screen (60 Hz frame rate) via a projector (Sanyo, PLC-XP-100L) and viewed through a mirror mounted on the head coil (distance between mirror and eyes: about 12 cm). The background of the screen was uniform gray. Stimulus presentation and response collection was controlled using ASF (*Schwarzbach, 2011*), a toolbox based on the Matlab Psychophysics toolbox (*Brainard, 1997*).

Each functional run consisted of 56 experimental trials (28 exemplars of the action categories performed by each of the two actors) and 18 null events (to enhance design efficiency) presented in a pseudorandomized order (preventing that the same action was shown in two consecutive trials, except during catch trials, see next paragraph). A trial consisted of the presentation of an action image for 1 s followed by 3 s of fixation. A null event consisted of the presentation of a fixation cross for 4 s. Within run 1–6, each possible combination of action types (28) x exemplars (12) was presented once, for a total of 336 trials. For runs 7–12, the randomization procedure was repeated such that each possible combination was presented another time. In this way, each participant saw each exemplar twice during the entire experiment (and thus each action was presented 24 times). A full balancing of all combinations of action, scene, and actor within each run was not possible with 28 actions, therefore the experiment was quasi-balanced: in each run, if actor one performed an action in scene A, actor two performed the same action in scene B, and vice versa.

To ensure that participants paid attention to the stimuli, we included seven (out of 63; 4.41%) catch trials in each run which consisted in the presentation of an image depicting the same action (but not the same exemplar) as the action presented in trial N-1 (e.g. *eating* an apple, actor A, scene A, followed by *eating* a sandwich, actor B, scene B). Participants were instructed to press a button with their right index finger whenever they detected an action repetition. Within the entire experimental session, all 28 actions could serve as catch trials and each action was selected randomly without replacement such that the same action could not be used as a catch trial within the same run. After a set of 4 runs all 28 actions were used as catch trial once, thus the selection process started from scratch. Catch trials were discarded from multivariate data analysis.

Before entering the scanner, participants received written instructions about their task and familiarized with the stimulus material for a couple of minutes. Next, participants carried out a short practice run to ensure that they properly understood the task.

## MRI data acquisition

Functional and structural data were acquired using a Siemens TIM Trio 3T MRI scanner. For the acquisition of the functional images, we used a T2*-weighted gradient EPI sequence. The repetition time (TR) was 2.5 s, the echo time (TE) was 30 milliseconds, the flip angle was 85°, the field of view was 192 × 192 mm, the matrix size was 64 × 64, and the voxel resolution was 3 × 3 × 3 mm. A total of 37 slices were acquired in ascending interleaved order. Each functional run lasted 5 min and 55 s and consisted of 142 volumes.

For the structural data, we used a T1*-weighted MPRAGE sequence (image size 256 × 256 × 176 voxels, voxel size 1 × 1 × 1 mm, TR 1.9 s, TE 3.03, flip angle 11), lasting 5 min and 35 s.

## MRI data preprocessing

Anatomical data were segmented using FreeSurfer (*Fischl et al., 1999*). Preprocessing of the functional data was carried out using SPM12 (http://www.fil.ion.ucl.ac.uk/spm/software/spm12/). The slices of each functional volume were slice time corrected and then spatially realigned to correct for head movements. Functional volumes were then coregistered to the individual anatomical image. Analyses were conducted in individual volume space, but using the inner and outer surfaces obtained with FreeSurfer as a constraint to select the voxels included in each searchlight as implemented in CoSMoMVPA (*Oosterhof et al., 2011*; *Oosterhof et al., 2016*). The resulting maps were resampled to the surface level on the Human Connectome Project common space *FS_LR 32 k* (*Glasser et al., 2013*) using FreeSurfer and workbench connectome (*Marcus et al., 2011*). Multivariate analyses were conducted using unsmoothed data.

## Behavioral experiment

Following the fMRI experiment, either on the same or the next day, participants took part in an additional behavioral experiment in which they carried out an inverse MDS task using similar procedures as described above (see Materials and methods section and *Figure 1B*). The actions were similar to those used during the fMRI experiment. In separate blocks of the experiment, participants were asked to arrange the actions according to their perceived similarity in terms of (a) meaning (referred to as 'semantics' in the remainder of the text), (b) the body part(s) involved, (c) scene, (d) movement kinematics, (e) the object involved.

The order of blocks was counterbalanced across participants. Participants were provided with the following written instructions:

- In the semantic similarity task, you will be asked to arrange the images with respect of their meaning: for example, sewing and ironing should be placed closer to each other than sewing and smoking.
- In the body parts task, actions typically involving the same/similar body parts, for example kicking a ball and walking, should be placed closer to each other than kicking a ball and smiling.
- In the context similarity task, actions typically taking place in the same/similar context, for example cutting bread and preparing tea, should be placed closer to each other than cutting bread and cutting hair.
- In the type of movement task, you will be asked to arrange the images with respect to the type of movement usually involved in each action. For example, actions like grasping and reaching would be more close to each other than grasping and kicking.
- In the type of object task, actions involving similar objects, for example catching a football and throwing a tennis ball, should be placed closer to each other than throwing a tennis ball and throwing a pillow at another person.

To further characterize the structure that emerged from the inverse MDS, we adopted principal component analysis (PCA) as implemented in the R package *cluster* to individuate the principal components along which the actions were organized. To characterize the observed clusters, we furthermore used a model-based approach using the *K-means* (*Hartigan and Wang, 1979*) clustering method. The K-means method requires the number of clusters as an input, which was one of the parameters we wished to estimate from the data. To this aim, we used the Silhouette method (*Rousseeuw, 1987*) as implemented in the R package *factoextra* to estimate the optimal number of

clusters. Specifically, this method provides an estimate of the averaged distance between clusters as a function of the number of clusters used and selects the value which provides the maximal distance.

## Construction of representational dissimilarity matrices (RDMs)

To construct RDMs for the semantic, body, scene, movement, and object model used in the behavioral experiment, we used the same procedure described in the section **Stimulus selection**, that is we determined the Euclidean distance between each pair of actions that resulted from inverse MDS, and normalized the dissimilarity matrices by dividing each value by the maximum value of each matrix. Individual dissimilarity matrices were used as a model for the multiple regression-based representational similarity analysis of fMRI data (see section *Representational Similarity Analysis*). We found significant (all p-values were smaller than p<0.0001 and survived false discovery rate correction) inter-observer correlations, that is the individual RDMs significantly correlated with the average RDMs of the remaining participants (mean leave-one-subject-out correlation coefficients [min – max individual correlation coefficients]; semantic model: 0.61 [0.46–0.78], body model: 0.57 [0.31–0.70]; scene model: 0.63 [0.40–0.78]; movement model: 0.47 [0.26–0.67]; object model: 0.51 [0.22–0.71]. Clusters obtained from the body, scene, movement, and object model using PCA can be found in *Figure 3—figure supplement 4A–D*.

To construct RDMs for the sociality and the transitivity models (*Wurm et al., 2017*), we conducted an online experiment using Qualtrics and Amazon Mechanical Turn in which we asked a separate group of N = 20 participants to judge on a Likert scale (1: not at all; 7: very much) the sociality (i.e. the interaction between the actors involved) and transitivity (i.e. the use of an object) of each action. To construct the RDM for the distance model, we asked another group of N = 20 participants to judge the distance (1: within reaching distance; 2: not within reaching distance, but <= 3 meters, 3:>3 meters) from the observer at which each action takes place. For each participant, we derived a RDM from the judgment scores and then averaged the individual RDMs for the sociality, transitivity and distance model.

The model controlling for the number of people depicted in the action was constructed by coding one person involved as 0 and two people involved as 1. The HMAX-C1 model (*Serre et al., 2007*) was derived similarly to *Connolly et al. (2012)* in the following way: we computed the C1 representation of each stimulus image (i.e. each exemplar of an action) and then averaged across the exemplar response vectors to obtain one C1 vector for each action. The 28 HMAX-C1 representations were then used to compute the RDM to be used as a predictor in the multiple regression RSA.

To examine correlations between the different model representational dissimilarity matrices, we computed correlations between each pairwise comparison of RDMs, both averaged across participants (*Figure 2—figure supplement 1A*) and separately for each participant (*Figure 2—figure supplement 1B,C*). As can be seen, not surprisingly, there are modest correlations between the different models (in particular when averaged across participants). However, as shown in the Materials and methods section on multiple regression RSA, the Variance Inflation Factor indicated a low risk of collinearity and thus justified the use of these models for multiple regression RSA.

## Representational Similarity Analysis (RSA)

The aim of the RSA was to individuate those brain regions that were best explained by the models obtained behaviorally and thus to infer the representational geometry that these areas encoded. We therefore conducted an RSA over the entire cortical surface using a searchlight approach (*Kriegeskorte et al., 2006*) at the individual brain space. Each searchlight consisted of 100 features (one central vertex + 99 neighbors) and was approximately 12 mm in radius. All multivariate analyses were carried out using custom written Matlab functions (available at: osf.io/cvrb2 ; *Tucciarelli, 2019*) and CoSMoMVPA (*Oosterhof et al., 2016*).

For the multivariate analysis, the design matrix consisted of 142 volumes X 28 predictors of interest (resulting from the 28 actions) plus nuisance predictors consisting of the catch trials, the parameters resulting from motion correction, and a constant term. Thus, for each participant, we obtained 28 beta maps at the volume level. We adopted two approaches for the representational similarity analysis, a standard and a multiple regression RSA approach.

## Standard RSA

First, to identify clusters in which the neural data reflected the dissimilarity pattern captured by the semantic model, we conducted a standard RSA. For this analysis, the beta maps were averaged across runs. For each searchlight, we derived the normalized RDM using squared Euclidian distance as distance metric (note that we chose this distance metric, rather than Spearman correlation, for a more straightforward comparison of the results of the standard and multiple regression RSA; see also next paragraph). For each searchlight, the RDM was correlated with the normalized RDM of the semantic model. The correlation values were assigned to the central node of each searchlight, thus leading to a correlation map for the semantic model, separately for each participant. The correlation maps of this first-level analysis were then resampled to the common space and Fisher transformed to normalize the distribution across participants to run a second-level analysis. Specifically, the correlation maps for the semantic model of all N participants were tested against zero using a one-tailed *t*-test at each vertex. The resulting t maps were corrected using a cluster-based nonparametric Monte Carlo permutation analysis (5000 iterations; initial threshold p<0.001 ; *Stelzer et al., 2013*).

## Multiple regression RSA

To determine clusters in which the neural RDM correlated with a given RDM (with a specific focus on the semantic model) while accounting for the remaining RDMs, we conducted a multiple regression RSA at each searchlight. For this analysis, the beta patterns were first normalized across images and then averaged across runs. Before computing the neural RDM, the betas of a searchlight were also normalized across features. Following the procedure used by *Bonner and Epstein (2018)*, the neural RDM for each searchlight was computed using the squared Euclidean distance. Using the squared Euclidian distance as distance metric for multiple regression RSA, which models distances from a brain RDM as linear combinations of the distances from a number of predictor RDMs, guarantees that the distance metric sums linearly.

As predictors of the multiple regression analysis, we used the RDMs described in the section *Construction of representational dissimilarity matrices* (semantic, body, movement, object, scene, sociality, transitivity, distance, 1 vs 2 People, HMAX-C1; see also *Figure 5*).

To estimate potential risks of collinearity, we computed the Variance Inflation Factor (VIF) for each participant to have a measure of the *inflation* of the estimated variance of the *ith* regression coefficient (computed as $1/(1-R^2_i)$, where *i* indicates a variable and $R^2$ is the coefficient of determination), assuming this coefficient being independent from the others. The VIFs were relatively small (average VIF semantic model: 1.61, body model: 1.37, scene model: 1.63, movement model: 1.35, object model: 1.38, sociality model: 1.54, transitivity model: 1.31, distance model: 1.22, 1 vs 2 People model: 1.37, HMAX-C1 model: 1.06), indicating a low risk of multicollinearity (*Mason et al., 2003*) and thus justifying the use of multiple regression RSA.

For each participant, the multiple regression-based RSA provided us with beta maps for each of the ten predictors that were then entered in a second-level (group) analysis to test the individual beta maps against zero. The procedure for multiple comparisons correction was the same as described in the section *Standard RSA*.

## Acknowledgements

This work was supported by Royal Holloway University of London and a grant awarded to Angelika Lingnau (Heisenberg-Professorship, German Research Foundation, Li 2840/1–1 and Li 2840/2–1). We are grateful to Niko Kriegeskorte and Marieke Mur for providing code for the inverse MDS experiment, to Beatrice Agostini for helping with stimulus creation, and to Nick Oosterhof for advice on data analysis.

## Additional information

### Funding

| Funder | Grant reference number | Author |
| --- | --- | --- |
| Deutsche Forschungsge-meinschaft | Li 2840/1-1 | Angelika Lingnau |
| Deutsche Forschungsge-meinschaft | Li 2840/2-1 | Angelika Lingnau |

The funders had no role in study design, data collection and interpretation, or the decision to submit the work for publication.

### Author contributions

Raffaele Tucciarelli, Data curation, Formal analysis, Validation, Investigation, Visualization, Methodology, Writing—original draft; Moritz Wurm, Conceptualization, Methodology, Writing—review and editing; Elisa Baccolo, Investigation, Methodology, Writing—review and editing; Angelika Lingnau, Conceptualization, Resources, Supervision, Funding acquisition, Validation, Writing—review and editing

### Author ORCIDs

Raffaele Tucciarelli (iD) https://orcid.org/0000-0002-0342-308X
Moritz Wurm (iD) https://orcid.org/0000-0003-4358-9815
Elisa Baccolo (iD) https://orcid.org/0000-0002-2527-1953
Angelika Lingnau (iD) https://orcid.org/0000-0001-8620-3009

### Ethics

Human subjects: All participants provided written informed consent in line with local ethics and MRI protocols. The study was approved by the Ethics Committee at the Department of Psychology, Royal Holloway University of London (REF 2015/088).

### Decision letter and Author response

Decision letter https://doi.org/10.7554/eLife.47686.sa1
Author response https://doi.org/10.7554/eLife.47686.sa2

## Additional files

### Supplementary files

• Supplementary file 1. Cluster table standard RSA (semantic model).Cluster table standard RSA (semantic model). List of clusters resulting from the standard RSA for the semantic model which survived correction for multiple comparisons (cluster p-value<0.05; see *Materials and methods* and *Figure 4*). Coordinates are in MNI space. Labels are based on MRI scans that originated from the OASIS project (http://www.oasis-brains.org/) and were provided by Neuromorphometrics, Inc (http://www.neuromorphometrics.com/) under academic subscription provided in SPM12 and Glasser's surface-based atlas (*Glasser et al., 2016*).

• Supplementary file 2. Cluster table multiple regression RSA.Cluster table multiple regression RSA. List of clusters resulting from the multiple regression RSA for the eight different models (semantic, body, movement, object, transitivity, distance, 1 vs 2 people, HMAX-C1) which survived correction for multiple comparisons (cluster p-value<0.05; see *Materials and methods* and *Figure 6* and *Figure 6—figure supplement 1*). Coordinates are in MNI space. Labels are based on MRI scans that originated from the OASIS project (http://www.oasis-brains.org/) and were provided by Neuromorphometrics, Inc (http://www.neuromorphometrics.com/) under academic subscription provided in SPM12 and Glasser's surface-based atlas (*Glasser et al., 2016*).

• Supplementary file 3. List of actions identified using the online survey.We initially kept actions that were mentioned by at least 20% of the participants. The final selected twenty-eight actions used for the study are the ones highlighted. See *Materials and methods* section for the procedure used to select these actions.

• Transparent reporting form

## Data availability

Data and codes have been archived at the Open Science Framework (https://osf.io/cvrb2/).

The following dataset was generated:

| Author(s) | Year | Dataset title | Dataset URL | Database and Identifier |
|---|---|---|---|---|
| Raffaele Tucciarelli | 2019 | RESPACT: The representational space of observed actions | https://osf.io/cvrb2/ | Open Science Framework, cvrb2 |

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
