## [Decision Letter]

**Acceptance summary:**

Many regions in the human brain have been implicated in the representation of observed actions, but the literature on action recognition has focused largely on a few classes of actions. Here, the authors seek to understand the representation of actions by studying how the common underlying aspects of many different actions are reflected in brain responses. Perhaps the best aspect of this paper is the large number of actions used as stimuli. The realistic visual variation in these actions grants the work an important increment in external validity compared to much past work. The authors also cope well with the complexity of their stimuli by considering multiple models of the underlying components (or features) of actions, which capture multiple ways that actions can be similar. These models do not exactly rule out alternative explanations for the results in the paper. Rather, the authors show how different aspects of observed actions (or visual features correlated with actions) are represented in partly overlapping regions of human lateral occipito-temporal cortex. The comparison with the other models makes the finding that semantic aspects of actions best capture responses in a particular parcel of left lateral occipito-temporal cortex interestingly specific and compelling.

**Decision letter after peer review:**

Thank you for submitting your article "The representational space of observed actions" for consideration by *eLife*. Your article has been reviewed by three peer reviewers, including Mark Lescroart as the Reviewing Editor, and the evaluation has been overseen by Richard Ivry as the Senior Editor. The following individual involved in review of your submission has agreed to reveal their identity: Alon Hafri.

The reviewers have discussed the reviews with one another and the Reviewing Editor has drafted this decision to help you prepare a revised submission.

Summary:

The goal of the present study was to find which areas of the brain contain a representational space that reflects the semantic similarity space of actions. Using a set of photographic images of 28 everyday actions, inverse MDS, and multiple regression RSA of fMRI data, the authors found that semantic similarity ratings were best matched to representations in LOTC and left pIPS, over and above other models related to body parts, objects, movement, and scene contexts. The authors propose that these regions may act as the input for how humans represent the meaning of actions.

This work is significant in that it is the first to show that a punctate and limited set of regions (LOTC and pIPS) may be the substrate for human representation of action meanings from visual content. The broad stimulus set also provides an important generalization and expansion of past work on action representation. Consequently, the reviewers were unanimous in their enthusiasm for the impact and general quality of the study. However, the reviewers also raised concerns that must be addressed before the paper can be accepted.

Major comments:

1) The authors acknowledge some confounds in their experiment: some actions were photographed nearer than others, and some actions included two bodies where most included only one. The reviewers support the inclusion of these stimuli in the interest of broad sampling. However, the authors should create models to assess the severity of these potential confounds. Specifically, the authors should include a one- vs. two-person model and a near vs. far model as sixth and seventh models in the multiple regression RSA analysis.

2) The authors should also consider at least one image-derived model to assure that any differences they measure in the brain are not due to low-level differences between their conditions. The authors should choose a model that provides a better proxy for V1 processing than does the pixel energy model they currently use (e.g. a Gist model, Gabor model, HMax, or the first layer of nearly any convolutional neural network). Use of a better low-level model is necessary both to assess correlations with the other models and to be included in the multiple regression RSA analysis. Particularly if the RDM-RDM correlations are small (see below), there is the potential that only one or two outlying conditions in low-level similarity space could affect the results.

3) The authors should provide more information about the subjects' instructions in their behavioral rating sessions. Specifically, the authors should clarify whether "meaning" was defined for participants, and whether (and which) examples were given. These instructions strongly influence the interpretation of the models used in the study.

4) The authors should indicate whether some or all of the subjects were naive to the purpose of the study. This is necessary because the models of interest are all based on subjective ratings that might be influenced by knowledge of the purpose of the experiment.

5) Probabilistic regions, or at least probabilistic centroids of regions, for hMT+, EBA, and V3A/B should be labeled on the flatmaps. Such labels are available from published atlases and other work and would require no additional data collection. Currently, anatomical details are only reported for maxima of activation in Supplementary file 3. Labels on the flatmaps will make it clearer how the extent of activation in each condition relates to hMT+, EBA, and V3A/B (and by extension, to OPA/TOS). This is important given the extent to which anatomy figures in the Discussion.

6) The authors should report how large the model-RDM-to-brain-RDM correlations are in individual subjects, or at least an average and range of these correlations across subjects. The authors should also include some visualization of the brain-derived RDM(s) for an area or areas of interest to facilitate qualitative comparison with the model RDMs in Figure 2. One of the hazards of RSA is that correlation can be strongly influenced by a few outlying data points, so if a given model only captures one or two cells of the RDM well, small but above-chance correlation values can still result. If the overall correlation between the model RDMs and the brain RDMs is high (say, over 0.4), this is not a concern. However, in many RSA studies, the correlations between model- and brain-derived RDMs is quite low (r < 0.05), which is in the range that a few outlying values could affect results. If this were true here, it would have consequences for the interpretation of the models, so some assurance is necessary that this is not the case.

7) The authors should report the searchlight radius.

8) Instead of correlation distance, the authors should use squared Euclidean distance as their RDM distance metric, as in (Bonner and Epstein, 2018; Khaligh-Razavi and Kriegeskorte, 2014). Since the distances from one brain RDM are modeled as linear combinations of the distances from a set of predictor RDMs, a distance metric that sums linearly is more in line with the modeling assumptions.

9) The reviewers do not object to use of static images. However, the authors should more explicitly justify the use of static images instead of videos, and discuss the consequences of not using moving stimuli. (This topic merits than the one sentence currently present around the third paragraph of the Discussion.)

10) A substantial portion of the Discussion is dedicated to the results in pIPS. This section is fairly long and strays from the data reported in the paper somewhat more than the other sections. Since the analysis is performed at a group level and there are no functional localizers reported, and since the location of OPA in particular can vary across individuals and is not tightly localized to the Temporal Occipital Sulcus (Dilks et al., 2013), it is difficult to say what the relation of the IPS result is as compared to OPA or other regions. This section should be re-written to more clearly indicate which parts are speculative, and potentially reduced in length.

11) Currently, discussion of the relationship between the recovered action similarity space and other work on this topic is confined to a few lines in the Discussion. The authors' point that their space depends on the actions they chose to include is well taken, but nonetheless the authors should discuss whether and how the dimensions of their similarity space relate to other hypotheses about the semantic structure of action representation. Work in language has thought at length on this issue (e.g. Pinker, 1989; Talmy, 1985; Kemmerer).

---

## [Author Response]

Major comments:1) The authors acknowledge some confounds in their experiment: some actions were photographed nearer than others, and some actions included two bodies where most included only one. The reviewers support the inclusion of these stimuli in the interest of broad sampling. However, the authors should create models to assess the severity of these potential confounds. Specifically, the authors should include a one- vs. two-person model and a near vs. far model as sixth and seventh models in the multiple regression RSA analysis.

We thank the reviewers for this comment and we agree that the severity of these potential confounds should be assessed. To this aim, we carried out the multiple regression RSA with a number of additional models, as suggested by the reviewers. To construct the model depicting the distance between the observer and the action, we carried out an additional online rating study in N = 20 participants, asking participants to judge the distance between the observer and the actor (1: within reaching distance, 2: not within reaching distance, but within 3 meters, 3: further away than 3 meters). Moreover, we added a model depicting the number of people present in the action (1 vs. 2). To be able to compare our results with a previous study reporting sensitivity of the dorsal and ventral portion of the LOTC to sociality and transitivity, respectively (as also suggested by one of the reviewers), we carried out an additional online rating study to construct these two models as well. Finally, as suggested in Point 2, we also added the HMAX-C1 model as a regressor to account for any low-level differences between the conditions.

The results of the multiple regression RSA using the additional models described above are shown in Figures 7 and Figure 6—figure supplement 1 of the revised manuscript. Importantly, whereas the multiple regression RSA reported in the original version of the manuscript revealed clusters for the semantic model in the left parietal and left and right lateral occipito-temporal areas, the new analysis, including models for sociality, transitivity, distance, number of people, and the HMAX C1 model, only revealed the cluster in the left LOTC. This suggests that the results observed in the left parietal and the right LOTC in the previous version of the manuscript were indeed due to some of the variability accounted for in the models that we added to the new multiple regression RSA. By contrast, we can conclude with more confidence that the neural dissimilarity structure obtained in the left LOTC captured the semantic dissimilarity structure obtained behaviorally, over and beyond the dissimilarity captured in the nine other models.

Note that in contrast to the previous analysis, the multiple regression analysis for the scene model did not reveal any clusters that survived corrections for multiple comparisons, suggesting that the results we obtained for this model previously were accounted for in some of the models we added to the multiple regression RSA.

The transitivity model revealed a cluster that is in line with the cluster reported by Wurm, Caramazza, and Lingnau, 2017, whereas the sociality model did not reveal any clusters that survived corrections for multiple comparisons (note, however, that the current study was not designed to test this model, with only a small number of actions directed towards another person).

As can be seen in Figure 2—figure supplement 1 of the revised manuscript, there is a high correlation between the one- vs. two-person model and the sociality and transitivity models. However, the Variance Inflation Factor (VIF), reported in more detail in the manuscript, indicated a low risk of multicollinearity, thus justifying the use of a multiple regression approach.

As a consequence of these changes, we added the following paragraphs in the main manuscript:

Results section

“Given that a number of action components covary to some extent with semantic features (e.g. locomotion actions typically take place outdoors, cleaning-related actions involve certain objects, etc.; see also Figure 2—figure supplement 1), it is impossible to determine precisely what kind of information drove the RSA effects in the identified regions on the basis of the correlation-based RSA alone. […] Therefore, the multiple-regression RSA included ten predictors (semantic, body, scene, movement, object, sociality, transitivity, distance, 1 vs. 2 people, HMAX-C1).”

Materials and methods section:

“As predictors of the multiple-regression analysis, we used the RDMs described in the section Construction of representational dissimilarity matrices (semantic, body, movement, object, scene, sociality, transitivity, distance, 1 vs. 2 People, HMAX-C1; see also Figure 5).”

2) The authors should also consider at least one image-derived model to assure that any differences they measure in the brain are not due to low-level differences between their conditions. The authors should choose a model that provides a better proxy for V1 processing than does the pixel energy model they currently use (e.g. a Gist model, Gabor model, HMax, or the first layer of nearly any convolutional neural network). Use of a better low-level model is necessary both to assess correlations with the other models and to be included in the multiple regression RSA analysis. Particularly if the RDM-RDM correlations are small (see below), there is the potential that only one or two outlying conditions in low-level similarity space could affect the results.

We thank the reviewers for raising this point. As suggested, we constructed a model for early visual areas using the HMAX C1 model and added it as a regressor to the multiple regression RSA (see also response to point 1 above, and Figures 7 and Figure 6—figure supplement 1 of the revised manuscript). The correlations between the HMAX C1 model and the remaining models are shown in Figure 2—figure supplement 1 of the revised manuscript. Importantly, the HMAX C1 model showed no systematic correlations with the remaining models.

As a consequence of these changes, we updated the manuscript accordingly in the following sections:

Results section:

Original version:

“in bilateral anterior LOTC at the junction to posterior middle temporal gyrus, right posterior superior temporal sulcus, and left pIPS.”

Revised version:

“in the left anterior LOTC at the junction to the posterior middle temporal gyrus”.

Discussion section:

Original version:

…“we showed that this semantic similarity structure of observed actions was best captured in the patterns of activation in the LOTC (bilaterally) and left posterior IPL.”

Revised version:

…“we showed that this semantic similarity structure of observed actions was best captured in the patterns of activation in the left LOTC”.

3) The authors should provide more information about the subjects' instructions in their behavioral rating sessions. Specifically, the authors should clarify whether "meaning" was defined for participants, and whether (and which) examples were given. These instructions strongly influence the interpretation of the models used in the study.

Participants received the following written instructions (and examples):

- “- In the semantic similarity task, you will be asked to arrange the images with respect of their meaning: for example, sewing and ironing should be placed closer to each other than sewing and smoking […] - In the type of object task, actions involving similar objects, e.g. catching a football and throwing a tennis ball, should be placed closer to each other than throwing a tennis ball and throwing a pillow at another person.”

We added these written instructions to the Materials and methods section.

4) The authors should indicate whether some or all of the subjects were naive to the purpose of the study. This is necessary because the models of interest are all based on subjective ratings that might be influenced by knowledge of the purpose of the experiment.

All participants but one (RT, one of the authors) were naïve to the purpose of the study. We added this information to the Materials and methods section.

5) Probabilistic regions, or at least probabilistic centroids of regions, for hMT+, EBA, and V3A/B should be labeled on the flatmaps. Such labels are available from published atlases and other work and would require no additional data collection. Currently, anatomical details are only reported for maxima of activation in Supplementary file 3. Labels on the flatmaps will make it clearer how the extent of activation in each condition relates to hMT+, EBA, and V3A/B (and by extension, to OPA/TOS). This is important given the extent to which anatomy figures in the Discussion.

We thank the reviewers for this suggestion. To provide labels for hMT+ and V3A/B, we added Figure 6—figure supplement 1 in which we superimposed the Glasser surface based atlas (Glasser et al., 2016) on top of the flat maps depicting the results of the multiple regression RSA. Labels for EBA are harder to establish since, to our knowledge, this area is not currently captured in any probabilistic atlas. However, for ease of interpretation of the results of the body model, we added Figure 6—figure supplement 2 in which we superimposed the coordinates of a number of previous studies that used specific EBA localizers on the flat maps showing the results of the multiple regression RSA for the body part model.

6) The authors should report how large the model-RDM-to-brain-RDM correlations are in individual subjects, or at least an average and range of these correlations across subjects. The authors should also include some visualization of the brain-derived RDM(s) for an area or areas of interest to facilitate qualitative comparison with the model RDMs in Figure 2. One of the hazards of RSA is that correlation can be strongly influenced by a few outlying data points, so if a given model only captures one or two cells of the RDM well, small but above-chance correlation values can still result. If the overall correlation between the model RDMs and the brain RDMs is high (say, over 0.4), this is not a concern. However, in many RSA studies, the correlations between model- and brain-derived RDMs is quite low (r < 0.05), which is in the range that a few outlying values could affect results. If this were true here, it would have consequences for the interpretation of the models, so some assurance is necessary that this is not the case.

We thank the reviewers for this suggestion. To evaluate the representational geometry of the semantic cluster, for each participant we selected the 100 voxels neighboring the maximum T-scores for the semantic models using the multiple regression RSA.

Next, for each participant, we computed the neural DSM and compared it with the model DSMs using Kendall’s tau rank correlation coefficient. This was the same procedure used for each searchlight as described in the Materials and methods section “Representational Similarity Analysis (RSA) – Standard RSA”. The correlation scores were averaged across the 20 participants, and results are shown in Figure 8A of the revised manuscript.

Several aspects are worth noting in Figure 8A. First, as expected, the bar plot confirmed that the semantic model is the model that best correlates with the neural RDM (shown in Figure 8B of the revised manuscript). Not surprisingly (and in line with the results shown in Figure 7), also some of the other models significantly correlated with the neural RDM in the left LOTC with the exception of the scene model, the sociality model, the object model and the HMAX-C1 model. As noted by the reviewers, the averaged correlations between neural and model RDMs were quite low (semantic model: 0.0645, distance model: 0.0608, 1 vs. 2 people model: 0.0563; all other models < 0.05). However, these were significantly greater than zero for 6 out of the 10 models (marked by an asterisk in Figure 8A). To rule out the possibility that the significant correlation between the neural DSM and the semantic DSM was due to only a few outliers, we conducted a classical multidimensional scaling (MDS) on the averaged neural DSM to visualize the representational geometry of the data. The result of this analysis is reported in Figure 8C, and the results of the k-means cluster analysis carried out on the behavioral data in Figure 3. For ease of comparison, actions were assigned to the same color code as the corresponding clusters identified from the behavioral analysis. As evident, the neural DSM of this region appears to encode a similar geometry as the one observed at the behavioral level for the semantic task. Similar results were obtained using hierarchical clustering analysis (see Figure 8D).

We have updated the manuscript in accordance with this new analysis:

“To have a better idea of the representational geometry encoded in the left LOTC cluster, we extracted the beta estimates associated with the 100 features neighboring the vertex with the highest T score in the cluster in the left LOTC revealed by the multiple regression RSA for the semantic model. […] To facilitate the comparison with Figure 3, we assigned actions with the same color code as the corresponding clusters identified from the behavioral analysis. A hierarchical clustering analysis showed a similar result (see Figure 8D).”

7) The authors should report the searchlight radius.

The radius of the searchlights was approximately 12 mm in radius on average. We now report this in the Materials and methods section:”… and was approximately 12 mm in radius”.

8) Instead of correlation distance, the authors should use squared Euclidean distance as their RDM distance metric, as in (Bonner and Epstein, 2018; Khaligh-Razavi and Kriegeskorte, 2014). Since the distances from one brain RDM are modeled as linear combinations of the distances from a set of predictor RDMs, a distance metric that sums linearly is more in line with the modeling assumptions.

We thank the reviewers for this important comment, and for suggesting a way to address this. After consulting the papers suggested by the reviewers, we re-ran the multiple regression RSA using the squared Euclidean distance as metric. The results are shown in Figure 6 (semantic model) and Figure 7 and Figure 6—figure supplement 1 (all remaining models) of the revised manuscript.

In general, results are qualitatively similar and the Squared Euclidean metric seems to have improved the estimation of the betas as noted by the larger T scores compared to the ones obtained when using Spearman correlation as distance metric. However, there are also important qualitative differences between the two approaches: the left IPL cluster and the more anterior STS cluster for the semantic model were not present anymore when using the squared Euclidean distance. Likewise, the cluster revealed for the scene model in the previous analysis was no longer present. For the body model, a new cluster in the right posterior LOTC survived corrections multiple-comparisons correction.

Note that in light of the additional models (and correspondingly, additional figures) we added to the manuscript, and given that the multiple regression RSA in the current study in our view really shows the key results, we decided to remove the results of the standard RSA for all models except for the semantic model (which we kept in for a direct comparison of the clusters revealed with and without controlling for the remaining models).

It is obvious from the references above (and from the comments of the reviewers) that it is more appropriate to use squared Euclidian distance as our RDM distance metric. By contrast, we are under the assumption that Spearman correlation should be the recommended RDM distance metric for the standard RSA. While the squared Euclidian distances should be linearly proportional to correlation distances (see also Bonner and Epstein, 2018), we wished to examine how the choice of the RDM distance metric (Spearman versus squared Euclidian distance) affects the results of the standard RSA. This comparison is shown in Author response image 1. As can be seen, the statistical map revealed by the standard RSA using the two distance metrics look very similar, but not identical. In particular, the standard RSA using squared Euclidian distance reveals additional clusters in left and right dorsal premotor cortex that are not revealed by the standard RSA using Spearman correlation. Moreover, Tvalues in posterior parietal cortex appear to be strongest in parietal cortex and somewhat weaker in the LOTC, whereas the opposite appears to be true for the standard RSA using the squared Euclidian distance. Given these differences, we decided to show the results of the standard RSA using the squared Euclidian distance in the manuscript since we were concerned that differences between the standard and the multiple regression RSA would otherwise reflect a difference between using the semantic model only (standard RSA, Spearman) versus controlling for the remaining models plus differences induced by using another distance metric (multiple regression RSA, squared Euclidian distance).

**Author response image 1. respfig1:** Comparison of the results of the standard RSA using Spearman correlation (top panel) and squared Euclidian distance (right panel) as distance metric.

9) The reviewers do not object to use of static images. However, the authors should more explicitly justify the use of static images instead of videos, and discuss the consequences of not using moving stimuli. (This topic merits than the one sentence currently present around the third paragraph of the Discussion.)

As suggested by the reviewers, we now explained why we used static images instead of videos (see Materials and methods section, “Stimulus selection”, last paragraph) and discuss the consequences of using static images instead of moving stimuli in the Discussion (fourth paragraph).

10) A substantial portion of the Discussion is dedicated to the results in pIPS. This section is fairly long and strays from the data reported in the paper somewhat more than the other sections. Since the analysis is performed at a group level and there are no functional localizers reported, and since the location of OPA in particular can vary across individuals and is not tightly localized to the Temporal Occipital Sulcus (Dilks et al., 2013), it is difficult to say what the relation of the IPS result is as compared to OPA or other regions. This section should be re-written to more clearly indicate which parts are speculative, and potentially reduced in length.

In light of the results revealed by the new analysis, we decided to delete this paragraph.

11) Currently, discussion of the relationship between the recovered action similarity space and other work on this topic is confined to a few lines in the Discussion. The authors' point that their space depends on the actions they chose to include is well taken, but nonetheless the authors should discuss whether and how the dimensions of their similarity space relate to other hypotheses about the semantic structure of action representation. Work in language has thought at length on this issue (e.g. Pinker, 1989; Talmy, 1985; Kemmerer).

We agree with the reviewers that this point should be made more explicit. We have added a new section on this aspect in the Discussion:

“To the best of our knowledge, the only study that explicitly examined dimensions underlying the organization of actions focused on tools and the way they are typically used. […] Based on the existing literature on semantic categories, we expect that future studies, using a similar approach as described in the current study with a wider range of actions, will reveal additional categories and overarching dimensions that show similarities, but are not necessarily identical to the semantic categories described in the literature so far.”